# Diversity of Antibiotic Resistance genes and Transfer Elements-Quantitative Monitoring (DARTE-QM): a method for detection of antimicrobial resistance in environmental samples

Schuyler D. Smith [1,2], Jinlyung Choi[2], Nicole Ricker[2,3], Fan Yang[2], Shannon Hinsa-Leasure [4], Michelle L. Soupir[2], Heather K. Allen[3] & Adina Howe [1,2 ✉]

Effective monitoring of antibiotic resistance genes and their dissemination in environmental ecosystems has been hindered by the cost and efficiency of methods available for the task. We developed the Diversity of Antibiotic Resistance genes and Transfer Elements-Quantitative Monitoring (DARTE-QM), a method implementing TruSeq high-throughput sequencing to simultaneously sequence thousands of antibiotic resistant gene targets representing a full-spectrum of antibiotic resistance classes common to environmental systems. In this study, we demonstrated DARTE-QM by screening 662 antibiotic resistance genes within complex environmental samples originated from manure, soil, and livestock feces, in addition to a mock-community reference to assess sensitivity and specificity. DARTE-QM offers a new approach to studying antibiotic resistance in environmental microbiomes, showing advantages in efficiency and the ability to scale for many samples. This method provides a means of data acquisition that will alleviate some of the obstacles that many researchers in this area currently face.

[1] Bioinformatics and Computational Biology Program, Iowa State University, 2014 Molecular Biology Building, Ames, IA 50011, USA. [2] Department of Agricultural and Biosystems Engineering, Iowa State University, 1340 Elings Hall, 605 Bissell Road, Ames, IA 50011, USA. [3] Food Safety and Enteric Pathogens Research Unit, ARS-USDA National Animal Disease Center, 1920 Dayton Ave, Ames, IA 50010, USA. [4] Department of Biology, Noyce Science Center, Grinnell College, 1116 Eighth Ave, Grinnell, IA 50112, USA. ✉email: adina@iastate.edu

The global spread of organisms possessing antimicrobial resistance (AMR), and their associated antibiotic-resistant genes (ARGs), has posed an increasing threat to the health of both humans and animals alike[1–3]. Characterization of the presence and abundance of ARGs, i.e. the resistome, in environmental microbiome samples has stood as a major challenge for researchers monitoring these events[4]. Such studies have been impeded by the broad diversity of the genes, their low presence in most natural environments, the difficulty of extracting DNA from microbes in those environments, and their association with mobile genetic elements accounting for approximately one-quarter of the genetic material in these microbiomes[5].

The genetic diversity of ARGs has made targeted sequencing approaches non-trivial and has led to the application of whole-genome shotgun metagenomic methods for the characterization resistomes[6]. This approach is dependent on the availability of a gene reference database to classify reads as ARGs sequences but does not require a priori knowledge of which genes constitute the resistome being investigated[7]. Despite being effective for the task, the cost per sample of employing metagenomic methods to elucidate resistomes often inhibits studies from large-scale analyses. This cost is driven by the necessity of needing to indiscriminately sequence all DNA within an environment, of which the resistome often comprises only a fraction of a percent. Therefore, without sufficient sequencing depth and coverage, ARGs may be either underrepresented or undetected[8]. Methods to enrich for ARGs in sequencing libraries would allow for increased information per sample[9].

In the effort of finding a more efficient means for screening ARGs DNA sequences, a method implementing a bait-and-capture system to identify specific ARG targets was developed[10]. This approach utilized streptavidin-coated magnetic beads to capture 80-mer bait sequences to target genes of interest. The bait-and-capture method has been well-suited for the characterization of low- and high-abundance ARGs, and has demonstrated the ability to differentiate resistomes from different sample sources[11]. Another method of targeted gene sequencing that has been used for ARG characterization involves custom primers for performing a PCR-based amplicon library preparation. This type of sequencing has been used extensively in microbiome studies for community profiling via bacterial 16S rRNA genes and combines barcoded adapters to differentiate hundreds of samples pooled in a single library preparation[12]. It has previously been limited in the number of primers that could be incorporated for a single library, but a more recent version of amplicon library preparation for multiplexed primers now exists and has been implemented for biomarker detection in clinical studies[13–16].

Our study demonstrates the usage of this multiplexed amplicon library preparation for the detection of ARGs in environmental samples. We have termed our method of implementing this technology the Diversity of Antibiotic Resistance genes and Transfer Elements-Quantitative Monitoring (DARTE-QM). Our study was designed to demonstrate that DARTE-QM offers practical application to ARG screening through its ability to simultaneously detect and quantify hundreds of ARGs residing in samples from various environments and that it can achieve high accuracy and sensitivity identifying ARG targets.

## Results

**Design of primers and samples**. DARTE-QM employed 796 primer pairs designed to target 67 antibiotic-resistant families and 662 ARGs, as well as a synthetic oligonucleotide reference sequence and the V4 region of the 16S rRNA gene, in a multiplexed amplicon library preparation (Supplementary Data 1). Subsequent paired-end sequencing of 150 base pair reads was conducted using the Illumina MiSeq platform (USDA, Ames, IA). To evaluate the results of DARTE-QM against a reference, we constructed a mock-community microbiome comprised of DNA extracted from 20 isolates (Supplementary Data 2) with completed genome sequences (Supplementary Data 3). For each of the mock-community libraries, we included varying concentrations of a synthetic oligonucleotide reference sequence to evaluate accuracy of quantification. We also examined how DARTE-QM was able to characterize resistomes associated with manure, swine fecal, and agricultural soil samples (Supplementary Data 4).

**Evaluation of DARTE-QM sequencing products**. The sequencing data produced via DARTE-QM was unique in its high level of heterogeneity, as compared to traditional amplicon data generated from a singular DNA primer (e.g., 16S SSU rRNA). On account of the numerous and diverse gene targets in the sequencing library, processing of DARTE-QM data required amendment of the traditional microbiome analysis pipelines (Fig. 1). After quality control and processing, 16 of the 18 samples from the mock community were retained for downstream analysis (2 samples removed for less than 5000 reads passing quality filters). Quality filtering also resulted in the removal of 38 of the 61 environmental samples due to sequencing coverage below 5000 reads, likely caused by PCR inhibitors common of manure and soil samples[17–19], leaving 39 samples, in total, to be used in the evaluation of DARTE-QM. The 16 mock-community samples yielded a mean of 192,415 reads per sample, and a mean of 44,440 reads able to be aligned to ARG references (Supplementary Data 5). In our environmental samples, across all sources, we observed a mean of 170,775 reads and a mean of 19,138 reads aligned to ARG references per sample. The overall recovery of targeted reads from the raw sequences for all 80 samples tested with DARTE-QM was approximately 34% when taking into account both the ARG targets as well as the 16S rRNA targets that were not reported in the analysis.

**DARTE-QM successfully amplified targeted genes with high specificity and sensitivity**. Success for DARTE-QM was evaluated on both specificity and sensitivity, determined using the mock-community microbiomes. Construction of the mock-community from DNA sourced from sequenced genomes allowed for comparison of a theoretical profile (Fig. 2a) to our experimental observations of ARGs in these samples. In the combined genomes of the mock community, ARGs comprised 0.03% (56 ARG targets) of the total genome by base-pair count. DARTE-QM was able to produce 55 of those 56 ARGs found in the mock-community reference genomes (specificity = 55/56 = 98.2%), consistently identifying them across all 16 samples (sensitivity = 902/952 = 94.7%) (Fig. 2b). Particular resistance families that were not successfully captured by DARTE-QM included those associated with the *acrA* subunit of multidrug efflux pump systems, as well as genes encoding for chloramphenicol resistance (e.g., *catA*). While overall, target relative abundances were observed to be similar compared to the theoretical estimates, the quantification of particular ARGs, such as transposon-associated *lnuC* conferring resistance to lincomycin, were found in higher abundance by DARTE-QM. Meanwhile, others such as *mecA* conferring methicillin-resistance were found to be underrepresented. There were also some ARGs detected in mock sample libraries that were not found within the mock draft genomes, these 36 ARGs represented 5.8% of all ARG reads from the mock-community samples. With regard to the synthetic oligonucleotide, there was a strong correlation observed reads (Supplementary Fig. 2, $R^2 = 0.91$) between the read abundance produced by DARTE-QM and the experimental concentration.

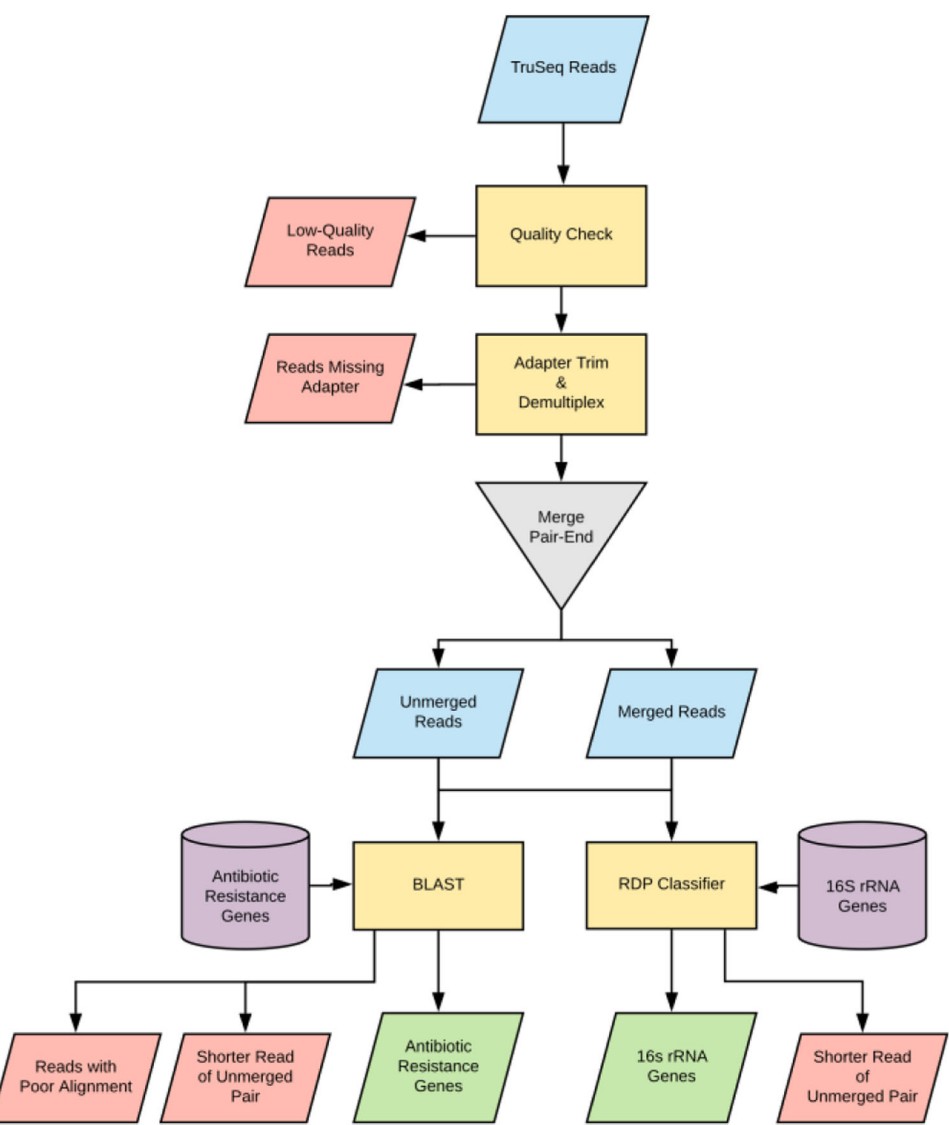

**Fig. 1 DARTE-QM bioinformatic pipeline.** Boxes colored blue represent reads retained through the pipeline, red were discarded, and green are the finalized reads ready for analysis. Reads were filtered by quality score and demultiplexed by the presence of primer sequences. To classify ARGs, both merged and unmerged reads were required to align to known genes in ResFam and CARD ARG reference databases. In the case of unmerged reads, if both the forward and the reverse read aligned to the same target, the shorter alignment from the pair was discarded.

We also observed a substantial number of reads in our sequencing datasets with primers located on the 5' end of the sequences but were unable to be aligned to any of our reference ARGs nor any position in the mock-community genomes. Inspection of a subset of these reads found that they contained repeated poly-A and poly-T elements. These reads were observed as unique sequences within the dataset, implying little or no biological pattern. These artifacts accounted for 47% of all reads in samples that passed quality controls. in samples that failed to pass quality filters, these sequences accounted for 85% of reads. The sample source appeared to be a significant, yet likely confounded, factor in the production of these artifacts. Sequencing of samples from the mock-community had significantly lower counts for artifact reads as compared to environmental samples (soil-A, $p = 0.038$; manure-A + soil-A, $p = 0.035$; swine fecal, $p < 0.001$, pairwise-Wilcoxon). Across all samples, an inverse linear relationship ($R^2 = 0.68$) (Supplementary Fig. 1) was observed between the number of reads which had a primer identified and the percentage of those reads with primers that were identified as artifacts.

**DARTE-QM differentiated resistomes between environmental sources**. DARTE-QM detected 240 ARG targets across all samples in this study (including 121 in Soil-A, 172 in Soil-B, 182 in Soil-C, 202 in Swine Manure-A, 129 in Swine Manure-B, 178 in the Swine Fecal samples, and 156 in the mock-community, Supplementary Data 6). Distinctions in the composition of resistomes were detected, not just from the presence of unique ARG targets but also from the abundance of the ARGs that composed the resistomes from each environment (Fig. 3). Ordination, via principal coordinate analysis based on Bray-Curtis distances of observed ARGs targets, showed clear separation of environmental sources, with the first two eigenvalues accounting for nearly 80% of the total variation (Fig. 4). Permutational multivariate analysis of variance (PERMANOVA) was used as a non-parametric multivariate statistical test to compare the variation of samples and environmental source. The results of the PERMANOVA test corroborated the apparent findings of the PCoA, and environmental sources were associated with a significant ($F = 11.45$, $R^2 = 0.70$, $p < 0.001$) portion of variation observed in the resistome profiles. DARTE-QM identified specific

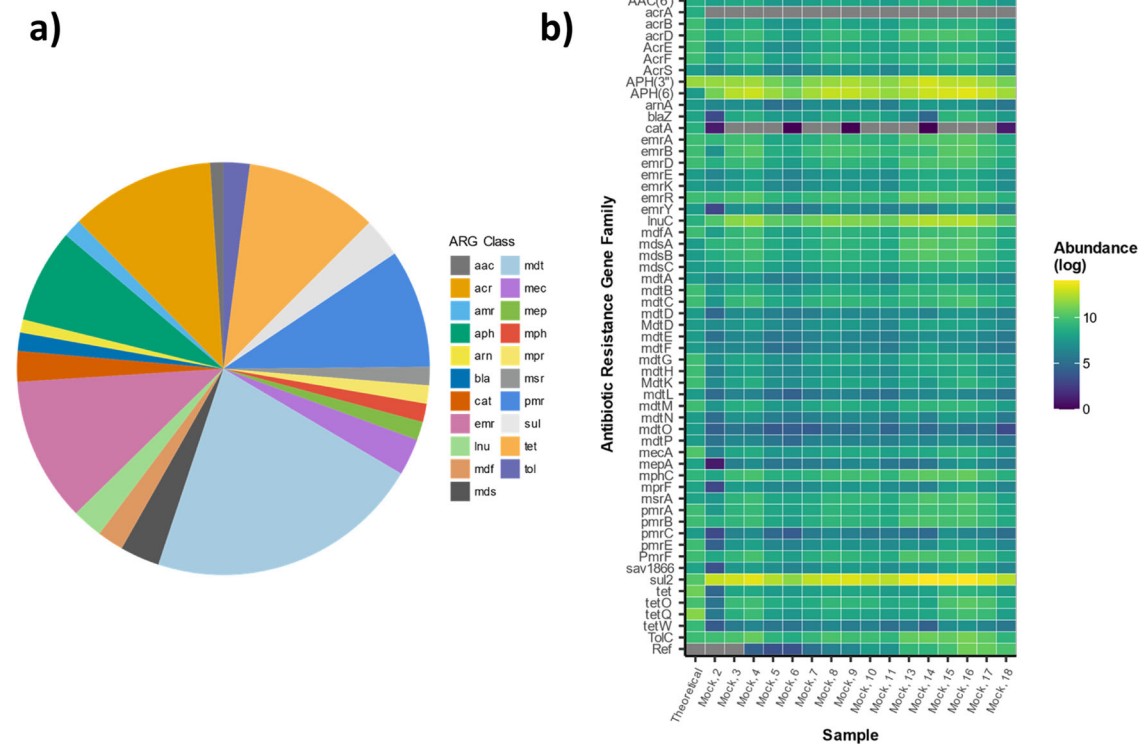

**Fig. 2 Presence and distribution of known ARGs within mock-community samples. a** The proportion of the resistome known to be represented by each ARG class based on sequenced genomes. **b** Heatmap depicting the log-transformed relative abundances of each ARG family from each mock sample and the theoretical distribution within genomes.

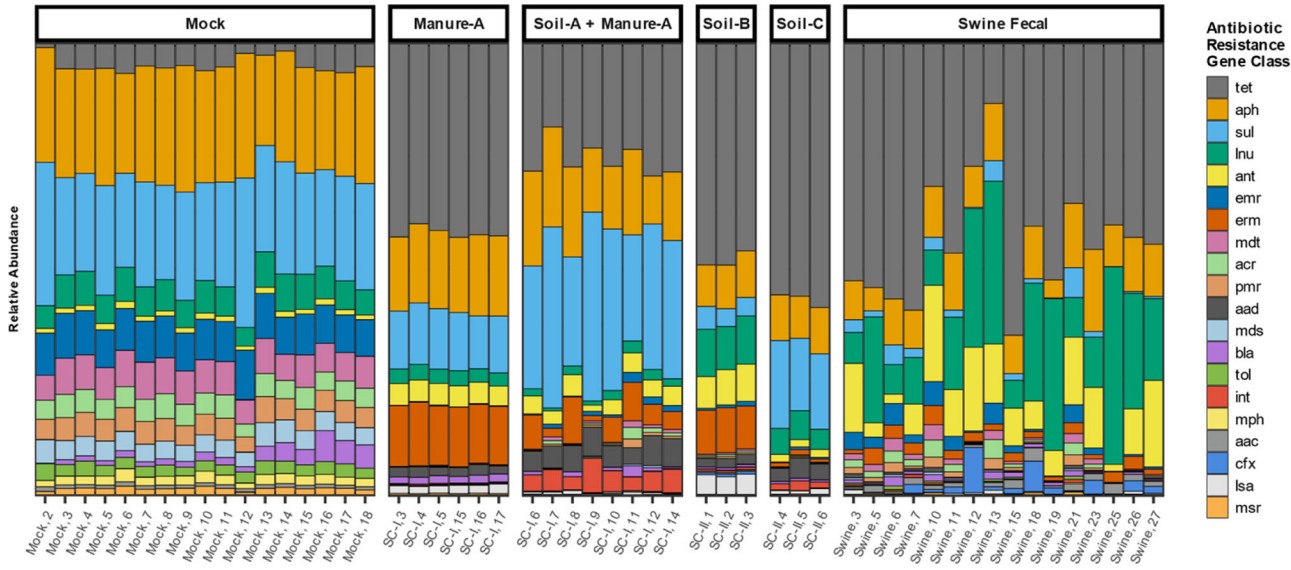

**Fig. 3 ARG profiles by source matrix.** Relative abundance of ARG classes identified for all mock community, swine fecal, soil, swine-manure, and manure-treated soils.

ARG patterns which distinguished resistomes sourced from different environmental samples, the most notable of which was within swine fecal samples where a distinctive presence of genes related to lincosamide and aminoglycoside resistance were observed. In the soils, with varied field management histories of swine and bovine manure amendment (soils B and C), we observed distinct characteristics of resistomes as well. Bovine manure-associated soils were found to be enriched with genes associated with resistance to aminoglycosides and sulfanomides, whereas the swine-manure-amended soils were replete with

aminoglycoside, lincosamides, and erythromycin-resistance-related genes.

**DARTE-QM produced results with comparable resolution to that of metagenomes.** Soil-column samples used in this study had been previously characterized through metagenome sequencing[20] (NCBI SRA Study SRP193066). DNA from the same sources were used for sequencing with DARTE-QM study for comparison of the two methods. Metagenomes from the soil samples had an average of 241 ARG reads and were excluded

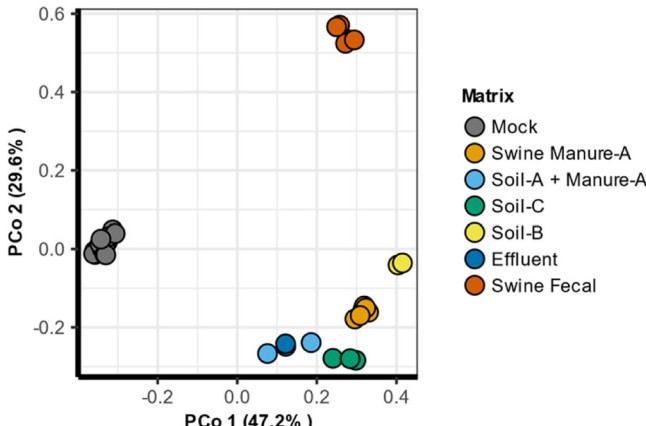

**Fig. 4 Ordination of DARTE-QM samples.** Principal coordinate analysis (PCoA) based on Bray–Curtis distances of the resistomes produced from DARTE-QM samples passing all QC-filtering.

from analysis; DARTE-QM returned a mean abundance of 5839 ARG reads in those same samples. Four swine-manure samples from the metagenome study yielded a mean ARG abundance of 76,226 reads, and the 12 manure-treated soil samples yielded an average of 7377 ARG reads. DARTE-QM produced mean abundances of 32,678 and 13,488 ARG reads in the same samples.

Relative abundance of ARG classes showed similar profiles for swine-manure from both technologies. DARTE-QM reads were classified into 99 ARG families and metagenome reads to 56 ARG families. From those, 39 ARG families were shared between the two methods and accounted for 89% and 84% of metagenome and DARTE-QM ARG families, respectively (Fig. 5a). In the manure-treated soil samples DARTE-QM identified 99 ARG families and metagenomes 92, sharing 50 of those that accounted for 90% and 83%, respectively. For exploratory identification of ARGs, DARTE-QM is disadvantaged by being a targeted method. For example, the metagenomes had a noticeable presence of genes from the AMR gene families for resistance-nodulation-cell division antibiotic efflux pump (*Mux* and *Mex* ARG Classes), which were not targeted by DARTE-QM. A direct comparison of both approaches constrains the metagenomes to those targeted by the primers of DARTE-QM (Fig. 5b). In this comparison, metagenomes identified 48 ARG families in the swine-manure samples and 65 in the manure-treated soils. Diversity measurements using the Shannon–Weiner Index of ARG classes showed similar values between the methods with DARTE-QM having $H = 2.95$ in swine-manure samples and $H = 2.87$ in manure-treated soil samples, while metagenomes had $H = 2.92$ in swine-manure samples and $H = 2.84$ in the manure-treated soils.

**DARTE-QM can distinguish gene variants through sequencing.** Two high-abundance genes, *erm35* encoding for the macrolide-lincosamide-streptogramin and *tetM* for tetracycline resistance, were selected for variant analysis. DARTE-QM reads classified as either of these genes were clustered at 97% nucleotide identity, resulting in three clusters for *erm35* and five clusters for *tetM*. Each cluster contained a minimum of ten unique sequences. The primary *erm35* cluster contained 4785 reads (Supplementary Data 7, Supplementary Fig. 3a). The other two *erm35* clusters were defined by 5–10 base pair variations within the associated 13 and 18 reads. Similarly, from a total of 24,653 reads classified as *tetM*, 96% defined the primary cluster, which was identical to one of the 6 *tetM* primer targets. Four of the other clusters, which contained between 32 and 676 reads, were defined by 9 and 24 base pair variations (Supplementary Data 7, Supplementary

Fig. 3b). Bacterial hosts associated with the observed *erm35* variants were solely associated with *Bacteroides coprosuis* and *Bacteroides spp*. and is consistent with the limited diversity of known isolates carrying this gene. In contrast, the sequences associated with *tetM* clusters are known to originate in various taxa. The largest *tetM* cluster was found to be highly conserved across a broad diversity of Gram-positive and some Gram-negative isolates. In comparison, the *tetM* cluster containing 676 reads, was primarily associated with plasmids found in *Escherichia coli* and *Salmonella*. The lower abundance of this cluster in the DARTE-QM data is consistent with the low relative abundance of Enterobacterales in swine gut-associated samples[21]. Similarly, the other *tetM* clusters were associated with *Streptococcus* strains and found in a lower diversity of taxa compared with the largest cluster.

## Discussion

DARTE-QM was conceptualized as an approach toward more efficient characterization of ARGs found in microbiomes. Specifically, we developed DARTE-QM to address the cost limitations of metagenomic approaches for ARG monitoring in environmental samples, where ARGs of interest often require significant sequencing depth and coverage for identification. One of the major goals was to be able to drastically scale the number of samples able to be evaluated by leveraging the high-throughput capabilities of barcode-multiplexing combined with amplicon library preparation. Similar to other amplicon-sequencing platforms, the costs of DARTE-QM are driven by the synthesis of primers and the price of sequencing. As DARTE-QM targets specified genes for amplification, it is able to enrich and detect ARGs that are present in low abundance, which is often a shortcoming for shotgun metagenomics. The number of samples that can be processed using DARTE-QM is limited by the number of unique barcode sequence adapters, the sequencing depth required per sample, and the number of gene targets. At the time of this study, the number of gene targets was constrained by the TruSeq platform, which currently supports 1536 primers and 96 barcoded samples.

The aim of this study was to demonstrate the efficacy of DARTE-QM for characterizing ARGs from environmental samples. Our results showed that DARTE-QM had success detecting the presence of hundreds of diverse ARGs across soil, manure, water, and our mock-community samples. While DARTE-QM was designed with the capacity to identify diverse ARG targets, our assessment was limited by ARGs contained in our samples. We used DNA extracted from isolates with sequenced genomes and ARG distributions to evaluate the sensitivity and accuracy of DARTE-QM. We observed strong performance for detecting ARGs in our mock community, having 98% of ARGs detected with high sensitivity and specificity.

One limitation of DARTE-QM is that the number of reads that were observed in the dataset but not able to be classified as one of the targeted ARGs. Many of these reads were characterized by the presence of a primer but with no sequence homology to a known sequence. While it is possible that these genes could be non-specific amplification of primers targeting other biological genes, the presence of poly-A and poly-T sequence patterns, like those seen in single-cell amplification[22], along with their majority singleton presence, suggested that they were sequencing artifacts. Though these artifacts present an impediment for leveraging the full sequencing coverage capability of DARTE-QM, we found that, with at least 25,000 reads per sample, we could identify 90% of the ARGs present in the mock-community samples. These sequencing artifacts also seemed to be produced by particular primers and in samples from specific environments, suggesting

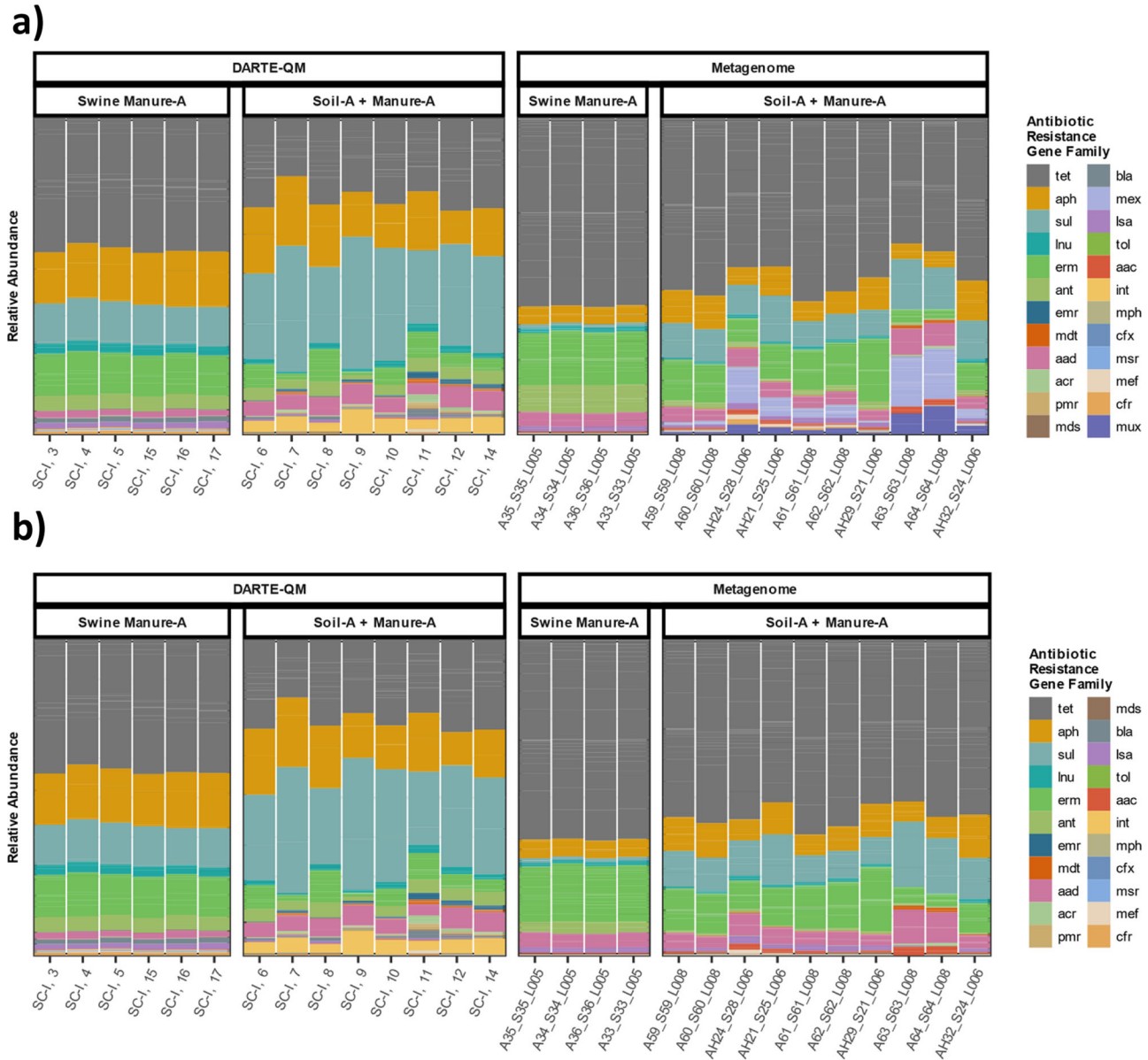

**Fig. 5 Comparison of ARGs captured by DARTE-QM and metagenome analyses. a** Distribution of all ARG families detected by DARTE-QM and metagenomes for the DNA originating from swine-manure (Swine Manure A) and swine-manure-amended soils (Soil-A + Manure-A). **b** Distribution of only ARG families targeted by DARTE-QM detected by both DARTE-QM and metagenomes for the same samples.

opportunities for optimization in future development of DARTE-QM. For instance, the primers targeting vancomycin-associated ARGs produced large number of artifact reads, and no vancomycin ARGs were expected in any of our samples. Another potential cause for the observation of non-targeted ARGs was the conservative parameters used for identifying a successful read (requiring an almost exact primer and ARG sequence match, primer error = 0.1 and percent identity = 98).

Generally, in samples where there was high-quality DNA and fewer unique ARGs (e.g., mock-community samples), DARTE-QM successfully characterized resistomes. Many of the samples that produced the highest percentage of reads as artifacts and/or failed quality control due to insufficient reads were from soils with a history of manure application or swine fecal samples (Supplementary Data 4), mediums known to have PCR inhibitors[23,24]. Artifacts can be filtered through target alignment and classification, with sufficient sequencing coverage, however, future studies aimed at improving the sequencing library

preparation and amplification for these sample types would be beneficial. In manure-associated samples, it is likely that the high organic matter content inhibited amplification, and modifications to DNA extraction protocols for its targeted removal should improve performance[25–27]. The DNA used in this study had previously been used for amplification of the 16 S SSU rRNA gene and/or metagenome libraries with success. Thus, it also likely that primer optimization would benefit detection of specific ARGs. In this application, DARTE-QM detects hundreds of ARGs simultaneously, increasing the probability for amplification artifacts[28,29]. For future studies, optimization for select gene targets would be recommended.

In cases where DARTE-QM abundances varied the most from the expected estimates, the gene targets were often associated with plasmids and other mobile elements. Multiple copies of these genes can exist for each individual cell, which would result in the genes being underestimated. For instance, *aph3-ib*, *aph6-id* and *sul2* are all found on the same IncQ plasmid. This is a likely

reason for the results of much higher observed copy numbers than other ARGs, as well as the theoretical estimate. The IncQ plasmid has been reported to have anywhere between 10 and 16 copies per cell.[30] The gene *aph(3′)-IIa*, is located on an IncI2 plasmid, which conversely is a low copy number plasmid[31], and is consistent with our results. Additionally, there were ARGs identified in mock samples that were not present in draft mock genomes. One explanation for these observations is the mis-assembly or absence of these genes in the draft mock genomes, as ARGs often contain repetitive elements and are difficult to assemble without sufficient coverage. Another possibility is related ARGs not being able to be distinguished by alignment, and the reads being misclassified as the similar gene.

The optimization of future versions of this platform for specific gene targets is possible. In the case of plasmid-associated genes or genes for which amplification failed, PCR conditions could be varied for optimal amplification and specific gene standards could be included for absolute quantification. Further, it is possible to select primers for DARTE-QM to target specific resistance classes, rather than the broad array of targets demonstrated in this study. In comparing DARTE-QM ARGs to those found in similar metagenomes, the ARG classes were found to be largely similar but relative abundances differed. These differences reflect contrast between the two technologies, with DARTE-QM determining relative abundances of ARGs within amplified targeted genes and metagenomes not biased by primer-specific targets. In other words, low abundant genes in metagenomes may be more difficult to detect in metagenomes because of the limits of detection of sequencing coverage and benefit with targeted amplification in DARTE-QM.

There was evidence of DARTE-QM's ability to quantify ARG presence using the correlation of read count to the varying concentrations of our synthetic oligonucleotide reference in the mock-community samples. Those results, though not a perfect correlation ($r^2 = 0.91$), illustrate that DARTE-QM is affected more by the amount of DNA available for the primer than by the competition between primers to find targets. Finally, comparisons to metagenomes suggested that DARTE-QM could detect similar measures of diversity of ARGs from samples. While the distributions of ARGs within the resistomes varied between DARTE-QM and metagenome resistomes, the differences between environmental sources could be distinguished, and broad patterns of resistance classes were similar. Combined, these results confirm that the primers used for DARTE-QM successfully amplified ARGs despite the potential for interference when simultaneously amplifying multiple gene targets in uniform conditions.

The most beneficial aspect of DARTE-QM to improving microbiome ARG monitoring is its ability to detect ARGs at costs that will allow hundreds of samples to be screened simultaneously. A current challenge to AMR monitoring is that characterizing broad indicators are expensive, and thus it is difficult to standardize studies for monitoring. DARTE-QM is a complement to existing approaches to characterize ARGs. We envision an optimal system whereby the most relevant ARGs in a study can be detected with less bias using metagenome sequencing, and these ARGs can subsequently be targeted for numerous samples using DARTE-QM. The sequencing from DARTE-QM can then provide information on the distribution of ARGs, as well as sequence variants, in a systematic fashion, even if in low abundance.

DARTE-QM represents the demonstration of simultaneous library preparation and subsequent sequencing of hundreds of unique gene targets from environmental DNA. Here, we demonstrated this application for the characterization of ARGs and associated resistomes in environmental samples, however,

DARTE-QM presents the opportunity to apply this approach toward gaining sequencing information for other diverse functional genes as well. This platform is particularly suited for studies in which genes of interest are numerous and well-defined, and where sequencing information from these genes would provide benefits to understanding biological operations (e.g., point mutations or association with sequences with host information). The ability to affordably scale for numerous genes and samples provides a much-needed resource for not only the field of AMR but for researchers interested in scaling functional gene characterization. Finally, we recognize that this is the first evaluation of DARTE-QM and that there are significant opportunities to further develop this approach for more targeted study. Given the simultaneous amplification of primers in DARTE-QM, we expect that the more specific the gene targets, the more optimized the library preparation can be for reliable quantification.

## Methods

**Sequencing targets and primer design**. Antibiotic resistance gene (ARG) targets for primer design were chosen and aggregated from two sources. There were 2472 sequences were obtained from the ResFinder database (version 3.2, November, 2016)[32], associated with 67 antibiotic resistance families. ResFinder was selected on account of its manual curation of genes associated with acquired antibiotic resistance. An additional 409 ARG-associated sequences chosen as well, which had previously demonstrated high prevalence in animal agriculture based on their relative abundance in previously published metagenomes[20]. To abide with the limitation of the number of allowed primers with the Illumina TruSeq library preparation, later described, the conglomerate of the chosen sequences was ultimately curated to representative sequences that targeted genes deemed of most interest to antibiotic resistance in agriculture. A single 300 bp synthetic oligonucleotide sequence was designed for use as a reference (reference target gene in Supplementary Data 1). The synthetic oligonucleotide was designed with no biological context to ensure that it would not interfere with any ARG detection, save for appropriate restriction sites that were added to allow for insertion into a pUC19 cloning vector. The sequence was compared to the entirety of the NCBI Genbank database and was confirmed to share no significant similarity to any existing records. Lastly, we included 25 sequences based on those used by the Earth Microbiome Project[33] to target the V4 variable region of the 16S rRNA gene.

The goal of primer design was to target the maximum number of our chosen sequences, with the highest specificity, staying within the set limit of 1536 primers for the library preparation. Primers were designed using the Ribosomal Database Project's EcoFunPrimer software:[34] product minimum length = 220, product maximum length = 330, Oligo minimum size = 22, oligo maximum size = 30, maximum mismatch = 0, temperature minimum = 55, temperature maximum = 63, hair-pin max = 24, homo-max = 35, assaymax = 30, degenmax = 6, noTEendfileter = T, nopoly3GCfilter = T, polyrunfilter = 4, GCfilter min = 0.15 GCfilter max = 0.8. This produced 1340 primers (Supplementary Data 1) to target the ARG-associated sequences, which accounted for 2184 sequences (88.3%) from those selected (Supplementary Data 8). Two primers were created for the synthetic oligonucleotide, and 30 were included for targeting all degeneracies of the 25 16S rRNA sequences. In total, DARTE-QM used 1372 primers (668 forward-primers, 704 reverse-primers) for 796 primer pairs to be used with Illumina's TruSeq Custom Amplicon Low Input library preparation. These primers targeted representative sequences of all 67 antibiotic-resistant families and 662 ARGs.

**Library Prep**. Oligonucleotide primers were created in Illumina Design Studio and ordered through Illumina (Supplementary Data 1). Paired-end libraries for each sample were prepared using the TruSeq Custom Amplicon Low Input Kit (Illumina) according to the manufacturer's instructions. This kit allows generation of up to 1536 amplicon targets over 96 samples. All DNA was diluted to 10 ng/µL during library preparation, or prepared with no dilution where concentrations were less than 10 ng/µL. An Agilent High Sensitivity D1000 ScreenTape System (Agilent Technologies) was used for measuring DNA concentration of prepared libraries. For sequencing, the MiSeq Reagent Kit v2 (300-cycles) (Illumina) reagents were used with the MiSeq sequencing platform.

**Mock-community**. A mock-community composed of 20 cultured isolates[35] was created for purpose of assessing the effectiveness of DARTE-QM. Nineteen of the genomes were available from the NCBI GenBank, and a single genome was sequenced at the USDA Animal Research (Ames, Iowa) (Supplementary Data 3). The ARGs found within the genomes were annotated using ResFam and also the Comprehensive Antibiotic Resistance Database (CARD, version 2.0.1)[36]. We included 6 mock-community samples sequenced in triple replicates with 0, 0.0025, 0.009, 0.025, 0.12, 0.25 ng of the synthetic oligonucleotide reference.

**Environmental samples**. To evaluate the practical implementation of DARTE-QM using environmental samples, we used 19 environmental samples originating from intrinsic and manure-amended soils, swine manures, effluent from manure-amended soils, and swine fecal samples that passed quality filters. Samples were selected from two previously published studies. In the first study, laboratory soil columns and rainfall simulations were used to evaluate the influence of swine-manure amendment on soils and effluent[20] (Supplementary Data 4). In the second study, fecal samples from swine with varying antibiotic usage and routes of administration were used[37]. Samples from a subsequent laboratory soil-column experiment designed to evaluate the influence of either swine or beef manure on soils and effluent were also included. Metagenomes are available for 14 samples (NCBI SRA database Bioproject PRJNA533779) and were used for comparisons to DARTE-QM results (10-30 million reads per sample).

**Data analysis**. All analysis was done in the statistical language R, unless otherwise stated. DARTE-QM sequences were quality checked using FastQC (v0.11.9)[38] (Fig. 1). Reads were demultiplexed by primer, which were identified and removed using Cutadapt (v2.10)[39] with an error tolerance of 0.1 and a Phred-score quality threshold of 20[39]. High-quality paired-end reads were merged using PEAR (v0.9.8)[40], requiring a minimum overlap of 10 bp. Merged reads were aligned against our database of targeted sequences using BLAST (v2.10)[41]. Successful alignment required a minimum of 90 bp and 98% similarity. For paired-end reads that were not able to be merged, each was aligned to the target database individually. If both reads aligned to the same target, the read with the longest alignment was selected as the representative sequence. We defined a successful amplification as a read for which a primer sequence was present, and the amplified sequence aligned to the primer's intended target with at least 90 bp length and at least 95% identity. Reads identified as having 16S rRNA primers were classified using the RDP Classifier[42] with default parameters, and then unpaired reads selected in the same manner as for ARGs.

The ability of DARTE-QM to quantify ARG presence was tested by comparing observed counts of the synthetic oligonucleotide to the expected concentrations. Samples were normalized by rarefying to a sequence count of 5000. Samples with a sequence count less than 5000 were discarded. Alpha diversity, richness, and ARGs was calculated using Shannon's index. Principal coordinate analysis was conducted to evaluate the variations of resistome profile in samples. Based on the relative abundance of ARGs in each sample, Bray-Curtis distances were calculated for each pair of samples, and the first two components of the eigenvalue decomposition were plotted. PERMANOVA was used to identify the significant factors (e.g., experiments, source-matrices) which contributed to the observed resistome variation. Cluster analysis was performed using k-means.

**Variant analysis**. To evaluate the presence of gene sequence variants, the observations of variants were estimated for sequences associated with the *erm35* and *tetM* genes. The forward reads of sequences that aligned to the DARTE-QM gene targets were clustered at 97% sequence similarity with CD-HIT (v4.6.7)[43]. Clusters containing greater than ten sequences were considered in our results, with representative sequences for each cluster determined by CD-HIT. Alignment was performed and visualized with JalView using ClustalW (v2.11.1.3)[44].

**Reporting summary**. Further information on research design is available in the Nature Research Reporting Summary linked to this article.

## Data availability

Sequence files, sample metadata, and the genome sequence for the mock-community member sequenced by the USDA facility in Ames, IA, can be found through FileShare with this link https://doi.org/10.25380/iastate.14390342

## Code availability

All code used for processing and analysis is open-source and can be found at https://schuyler-smith.github.io/DARTE-QM/ with a release archived at https://zenodo.org/record/5813982.

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

## Acknowledgements

This project was supported (or partially supported) by AFRI food safety grant no. 2016-68003-24604 from the USDA National Institute of Food and Agriculture. N.R. was supported by an appointment to the Agricultural Research Service (ARS) Research Participation Program administered by the Oak Ridge Institute for Science and Education (ORISE) through an interagency agreement between the U.S. Department of Energy (DOE) and the U.S. Department of Agriculture (USDA). ORISE is managed by ORAU under DOE contract number DE-AC05-06OR23100. This research was supported by appropriated funds from USDA-CRIS project 5030-31320-004-00D. We thank Jennifer Jones and Kathy Mou at the *ARS-USDA National Animal Disease Center* for their help with library preparation. We thank Jared Shelerud at Illumina for his help with the TruSeq Custom Amplicon platform.

## Author contributions

A.H., H.K.A., M.S., F.Y., N.R., and J.C. were designed the project; A.H. H.K.A., M.S., and S.H.-L. were involved in funding-acquisition; S.S, A.H., and N.R. analyzed the data and wrote the manuscript.

## Competing interests

The authors declare no competing interests.
