## [Transparent Peer Review File · Communications Biology]

Reviewers' comments:

Reviewer #1 (Remarks to the Author):

Smith et al. present a new method for the monitoring of antimicrobial resistance genes in the environment entitled DARTE-QM. The method is based on amplicon sequencing using primers designed to facilitate the amplification of AMR genes. The use of multiplexed amplicon library preparation at this scale is impressive on its own. Overall I believe this method to be a worthy addition to the environmental and medical microbiologists toolkit.

I have a few comments though:

line 73: "Despite being effective for the task, the cost per sample of employing metagenomic methods to elucidate resistomes often inhibits studies from scaling"

While the cost of shotgun metagenomics is often called prohibitive. But in fact, the cost for sequencing has dropped significantly over the recent years. Hence, I think this statement as well as the ones following about the costs of DARTE-QM requires quantification.

Line 76: "Therefore, it is often the case that only a minute subset of the sequencing-reads produced through this method will be informative to resistomes, and ARGs are either underrepresented or undetected, as sufficient sequencing depth and coverage is difficult to achieve."

This statement seems incongruent with the data shown later about the comparability between DARTE-QM and shotgun metagenomics where there are often more reads associated with ARGs found than using DARTE-QM. Further, it seems valuable to have the added information from shotgun metagenomic. It would be good to discuss how DARTE-QM compensates for the missing information (taxonomic, genomic context, etc).

line 121: "Quality filtering also resulted in the removal of 38 of the 61 environmental samples due to sequencing coverage below 5000 reads"

It is disconcerting to me that the method apparently fails for more than 50% of environmental samples. While this is quickly discussed by the authors, a longer discussion including possible remedies is warranted.

124: "The 16 mock-community samples yielded a mean of 192,415 reads per sample, and a mean of 44,440 reads able to be aligned to ARG references (Supp. Table 5). In our environmental samples, across all sources, we observed a mean of 170,775 reads and a mean of 19,138 reads aligned to ARG references per sample."

This is confusing to me. Does this imply that 75% to 90% of the amplicon reads are non-specific? How does this influence the results and how does this compare to typical amplicon experiments (for example for 16S rRNA)

Line 134: The definitions of true/false positive/negative seem to be slightly arbitrary and certainly suffer from information leakage. I don't think it is easily possible to define True Negatives in a way that is both correct and meaningful for this approach.

True Positives seems to be defined in a good way. False Positives represent off-targets of the primers, while False Negatives represent reads that either do not include the primer due to sequencing artifacts or random sequences of the correct genes (which should be extremely rare in such an experiment). I do not completely grasp the definition of True Negatives (reads within a sample assigned as TP outside of the primer in question??), but this definition seems to boost accuracy statistics due to the unbalanced nature of the sequencing set (i.e. the high amount of non-specific amplicon sequences). Further, there is significant information leakage as the reads are defined according to their sequences

which are also used as ground truth.
Please redo the derived statistics at the end of the paragraph accordingly.

Line 147: "Inspection of a subset of these reads found that they contained repeated poly-A and poly-T elements."

Do the authors have a more mechanistic explanation for this? Could the primer set be adjusted to minimize these effects? (I am not asking the authors to change the primer set, but discuss possible improvements)

line 162: "DARTE-QM was able to produce 55 of those 56 ARGs found in the mock-community reference genomes, consistently identifying them across all 16 samples"

How many ARGs that were not supposed to be found in the mock community samples were found?

line 209: "For identifying diverse ARGs, DARTE-QM is disadvantaged by being a targeted method."

Figure 4: While the ARG classes found using both methods appear to be largely similar, the ARG abundance patterns seem to be quite different. This should be discussed in more depth within the text. For example: Why are the sul and aph families detected in so much higher proportions using DARTE-QM?

Reviewer #2 (Remarks to the Author):

This paper present results on the development of a targeted approach to sequencing the resistome. In this paper, the authors develop a primer set targeting 662 antibiotic resistance genes called DARTE-QM. They verify the bait set on a mock community, and a variety of environmental samples. The authors provide valid statistical analysis. The quantitative aspect of this targeted approach is interesting and unique, as many other targeted approaches to studying the resistome are not quantitative (Guiton et al.). Unfortunately, the authors approach has several limitations. DARTE-QM failed to work in over half of the environmental samples tested, which is concerning for researchers. Furthermore, although the authors provided in vitro testing of their technique, they failed to validate it in silico. An in silico validation would help verify that DARTE-QM can capture all 662 antibiotic resistance genes, which the authors were unable to show in their in vitro approach. I recommend this manuscript for publication with major revisions. Overall, I really enjoyed reading this manuscript and with a few changes I think this manuscript will be much stronger.

Here are my specific comments:

Introduction:

Lines 74: I suggest instead of 'scaling' to try 'large scale'.

Line 76: I suggest replacing "minute" with "rare" or "small"

Lines 80-82: Another potential citation would be Beaudry et al., 2021 preprint on developing baits for AMR. Similar to the Guiton paper.

Results:

Lines 103-105: How did the authors determine what ARGs to target? How did the authors obtain the ARG sequences? Please provide a reference or reference the specific section in the materials and methods.

Lines 112-113: Can the authors explain what they mean by “true environmental resistomes”? Arguably, there is no such thing due to biases present in every step of the process (sampling, DNA extraction, PCR, sequencing, bioinformatics). This is why we use mock communities to establish ground truth. Please alter this sentence.

Lines 121-123: It is very concerning that the authors had to throw out over half of their samples. It seems like this method will only work for high quality samples. Can the authors address this in the discussion? Were they able to try anything to get these samples to work?

Lines 129-137: I think it would be helpful if the authors made a table for this information.

Lines 150-154: Why are so many reads off-target?

Figures:

Figure 2: I like the figure, however I think it is more intuitive to use a red-yellow-green scale rather than purple-green-yellow.

Figure 3. For the barplots they are separated by sample type. Why include this information on the x-axis also? It makes it harder to read – just include the sample ID number.

Figure 4. Same comment as above regarding the x-axis.

Discussion:

Lines 271-274: I think the authors need to be more specific here. In the materials and methods over half of the environmental samples did not work. That should be restated in the discussion, and this sentence should be altered to reflect that.

Lines 288-306: I think this paragraph should go third in the discussion. If you alter the last sentence of the second paragraph, like suggested above, it will flow nicely.

Other:

One thing that I think is lacking is an in-silico simulation of the authors probe set. They are testing these in “real world” and mock communities, but do not have an understanding of if they all will work in ideal wet lab conditions.

Could this approach be used in conjunction with other targeted approaches? Similar to how the authors suggest using it in approach with metagenomic libraries?

Do the authors want to compare and contrast the results from their study to other published studies on targeted approaches to sequencing the resistome?

We thank the reviewers for their comments and suggestions. Below, we respond to each comment with a response. We have enumerated each comment to help structure this response. The line numbers provided are referenced to the line-view in Word with "Simple" mark up in review (as we found that the line numbers would shift in other views between different systems). We have also attached a PDF-rendering of the line numbered document for reference.

Reviewer #1 (Remarks to the Author):

Smith et al. present a new method for the monitoring of antimicrobial resistance genes in the environment entitled DARTE-QM. The method is based on amplicon sequencing using primers designed to facilitate the amplification of AMR genes. The use of multiplexed amplicon library preparation at this scale is impressive on its own. Overall, I believe this method to be a worthy addition to the environmental and medical microbiologists toolkit.

I have a few comments though:

1. line 73: "Despite being effective for the task, the cost per sample of employing metagenomic methods to elucidate resistomes often inhibits studies from scaling"

While the cost of shotgun metagenomics is often called prohibitive. But in fact, the cost for sequencing has dropped significantly over the recent years. Hence, I think this statement as well as the ones following about the costs of DARTE-QM requires quantification.

Response:

Thank you for this comment to clarify the costs of metagenome sequencing and DARTE-QM. We agree that the cost of sequencing is decreasing and will continue to drop significantly. We are cautious to put specific costs in the text, given how rapidly they are changing. Instead, we have added why shotgun sequencing scales poorly for numerous samples, mainly the need to sequence at a depth great enough to get at ARGs that comprise only a fraction of the metagenome.

Line 75: "This cost is driven by the necessity of having to indiscriminately sequence all DNA within an environment, of which the resistome often comprises only a fraction of a percent. Therefore, without sufficient sequencing depth and coverage, ARGs may be either underrepresented or undetected⁸. Methods to enrich for ARGs in sequencing libraries would allow for increased information per sample⁹."

2. Line 76: "Therefore, it is often the case that only a minute subset of the sequencing-reads produced through this method will be informative to resistomes, and ARGs are either underrepresented or undetected, as sufficient sequencing depth and coverage is difficult to achieve."

This statement seems incongruent with the data shown later about the comparability between DARTE-QM and shotgun metagenomics where there are often more reads associated with ARGs found than using DARTE-QM. Further, it seems valuable to have the added information from shotgun metagenomic. It would be good to discuss how DARTE-QM compensates for the missing information (taxonomic, genomic context, etc).

Response:

Thank you, we have clarified this sentence, also in response to the comment above. We have added more specificity to the language to point out that resistomes can be characterized in metagenomes with sufficient sequencing coverage and also added summary info on the sequencing coverage used for the comparative metagenomes used in this study.

Line 75: This cost is driven by the necessity of having to indiscriminately sequence all DNA within an environment, of which the resistome often comprises only a fraction of a percent. Therefore, without sufficient sequencing depth and coverage, ARGs may be either underrepresented or undetected⁸.

Line 512: “Metagenomes are available for 14 samples (NCBI SRA database Bioproject PRJNA533779) and were used for comparisons to DARTE-QM results (10-30 million reads per sample).”

3. line 121: "Quality filtering also resulted in the removal of 38 of the 61 environmental samples due to sequencing coverage below 5000 reads"

It is disconcerting to me that the method apparently fails for more than 50% of environmental samples. While this is quickly discussed by the authors, a longer discussion including possible remedies is warranted.

Response:

Thank you for this point. We have added discussion on this topic, including citing literature where inhibitors of PCR have been challenging to molecular approaches in manure-associated samples and identifying methods that may help with future library preparation based on other studies.

Line 313: “Generally, in samples where there was high-quality DNA and fewer unique ARGs (e.g., mock-community samples), DARTE-QM successfully characterized resistomes. Many of the samples that produced the highest percentage of reads as artifacts and/or failed quality control due to insufficient reads were from soils with a history of manure application or swine fecal samples (Supp. Table 4), mediums known to have PCR inhibitors^{23,24}. Artifacts can be filtered through target alignment and classification, with sufficient sequencing coverage, however, future studies aimed at improving the sequencing library preparation and amplification for these sample types would be beneficial. In manure-associated samples, it is likely that the high organic matter content inhibited amplification, and modifications to DNA extraction protocols for its targeted removal should improve performance²⁵⁻²⁷. The DNA used in this study had previously been used for amplification of the 16S SSU rRNA gene and/or metagenome libraries with success. Thus, it is also likely that primer optimization would benefit detection of specific ARGs. In this application, DARTE-QM detects hundreds of ARGs simultaneously, increasing the probability for amplification artifacts^{28,29}. For future studies, optimization for select gene targets would be recommended.”

4. 124: "The 16 mock-community samples yielded a mean of 192,415 reads per sample, and a mean of 44,440 reads able to be aligned to ARG references (Supp. Table 5). In our environmental samples, across all sources, we observed a mean of 170,775 reads and a mean of 19,138 reads aligned to ARG

references per sample."

This is confusing to me. Does this imply that 75% to 90% of the amplicon reads are non-specific? How does this influence the results and how does this compare to typical amplicon experiments (for example for 16S rRNA)

Response:

The reviewer is correct, and the majority of DARTE-QM reads were observed to be non-specific to targeted ARGs. There are a few potential reasons that we can see. First, a portion of these reads are SSU amplicons (Supp Table 5). Second, these reads may represent the presence of biased PCR amplification and amplicon artifacts (discussion Lines 269 – 276). Finally, our requirements for correctly identified sequences were very strict, needing 98% identity. In typical amplicon experiments, the fraction of reads that cannot be dereplicated (in our hands) is usually less than 5%. In DARTE-QM, we are amplifying hundreds of probes at once rather than a single universal target (e.g., the V4 region of the SSU). Thus, it is likely that these simultaneous amplifications are causing bias and artifacts (discussed in Lines 310-333).

Line 293: "One limitation of DARTE-QM is that the number of reads that were observed in the dataset but not able to be classified as one of the targeted ARGs. Many of these reads were characterized by the presence of a primer but with no sequence homology to a known sequence. While it is possible that these genes could be non-specific amplification of primers targeting other biological genes, the presence of poly-A and poly-T sequence patterns, like those seen in single cell amplification²², along with their majority singleton presence, suggested that they were sequencing artifacts. Though these artifacts present an impediment for leveraging the full sequencing coverage capability of DARTE-QM, we found that, with at least 25,000 reads per sample, we could identify 90% of the ARGs present in the mock-community samples. These sequencing artifacts also seemed to be produced by particular primers and in samples from specific environments, suggesting opportunities for optimization in future development of DARTE-QM. For instance, the primers targeting vancomycin-associated ARGs produced large number of artifact reads, and no vancomycin ARGs were expected in any of our samples. Another potential cause for the observation of non-targeted ARGs were the conservative parameters used for identifying a successful read (requiring an almost exact primer and ARG sequence match, primer error = 0.1 and percent identity = 98)."

Also see addition included above to response #3, which discusses the role of PCR inhibitors which can result in reads not matching gene targets.

5. Line 134: The definitions of true/false positive/negative seem to be slightly arbitrary and certainly suffer from information leakage. I don't think it is easily possible to define True Negatives in a way that is both correct and meaningful for this approach.

True Positives seems to be defined in a good way. False Positives represent off-targets of the primers, while False Negatives represent reads that either do not include the primer due to sequencing artifacts or random sequences of the correct genes (which should be extremely rare in such an experiment). I do not completely grasp the definition of True Negatives (reads within a sample assigned as TP outside of the primer in question??), but this definition seems to boost accuracy statistics due to the unbalanced nature of the sequencing set (i.e. the high amount of non-specific amplicon sequences).

Further, there is significant information leakage as the reads are defined according to their sequences

which are also used as ground truth.

Please redo the derived statistics at the end of the paragraph accordingly.

Response:

Thank you for this feedback. We had used an analogous system of evaluation and can appreciate that it did not quite fit. We redefined the terms to be more transparent: reads that matched the primers, or not, and reads without primers (Supplemental table 6). We removed the TP FP TN FN definitions for sensitivity and specificity and used the simpler forms found in other TruSeq studies with the expected genes in the mock-community, how many were found, and how often.

Line 145: “Success for DARTE-QM was evaluated on both specificity and sensitivity were determined using the mock-community microbiomes. Construction of the mock-community from DNA sourced from sequenced genomes allowed for comparison of a theoretical profile to our experimental observations of ARGs in these samples. In the combined genomes of the mock-community, ARGs comprised 0.03% (56 ARG targets) of the total genome by base pair count. DARTE-QM was able to produce 55 of those 56 ARGs found in the mock-community reference genomes (specificity = $55/56 = 98.2\%$), consistently identifying them across all 16 samples (sensitivity = $902/952 = 94.7\%$) (Figure 2).”

6. Line 147: "Inspection of a subset of these reads found that they contained repeated poly-A and poly-T elements."

Do the authors have a more mechanistic explanation for this? Could the primer set be adjusted to minimize these effects? (I am not asking the authors to change the primer set, but discuss possible improvements)

Thank you for this comment. Since the completion of this study, we are now working on doing just this, improving primer amplification for specific gene targets (mainly the *tet* genes) and the results seem promising. We discuss possible improvements as suggested in more detail (see Response #3 and #4 for modified text).

7. line 162: "DARTE-QM was able to produce 55 of those 56 ARGs found in the mock-community reference genomes, consistently identifying them across all 16 samples"

How many ARGs that were not supposed to be found in the mock-community samples were found?

Response:

There were some ARGs that were found in the mock-community that were not known to be there based on the screening of the sequenced genomes. In some cases, those genes were related to known targets. These ARGs detected represented 5.8% of all ARG reads in the mock-community samples. The origin of these genes requires further study. It is likely they originate from the mock genomes, as many of these genomes are ‘draft’ quality and thus have not had the additional sequencing to assemble these ARGs, which have repetitive elements that are difficult to assemble.

Line 145 (Results): “There were also some ARGs detected in mock sample libraries that were not found within the mock draft genomes, these 36 ARGs represented 5.8% of all ARG reads from the mock-community samples.”

Line 345 (Discussion): “Additionally, there were ARGs identified in mock samples that were not present in draft mock genomes. One explanation for these observations is the mis-assembly or absence of these genes in the draft mock genomes, as ARGs often contain repetitive elements and are difficult to assemble without sufficient coverage.”

8. line 209: "For identifying diverse ARGs, DARTE-QM is disadvantaged by being a targeted method."

Response:

Thank you for this comment, and we have changed the wording of this statement.

Line 233: “For exploratory identification of ARGs, DARTE-QM is disadvantaged by being a targeted method.”

9. Figure 4: While the ARG classes found using both methods appear to be largely similar, the ARG abundance patterns seem to be quite different. This should be discussed in more depth within the text. For example: Why are the *sul* and *aph* families detected in so much higher proportions using DARTE-QM?

Response:

Thank you for this question. We believe that in the metagenomes, most of the reads will go to the most abundant genes (*tet* and *erm*) and saturate the detection. With the primer amplifications of DARTE-QM, we still see high abundance of these targets but we also amplify a greater diversity of other lower abundant ARGs. We have added to the discussion more details on these differences.

Line 355: “In comparing DARTE-QM ARGs to those found in similar metagenomes, the ARG classes were found to be largely similar but relative abundances differed. These differences reflect contrast between the two technologies, with DARTE-QM determining relative abundances of ARGs within amplified targeted genes and metagenomes not biased by primer-specific targets. In other words, low abundant genes in metagenomes may be more difficult to detect in metagenomes because of the limits of detection of sequencing coverage and benefit with targeted amplification in DARTE-QM.”

Reviewer #2 (Remarks to the Author):

This paper present results on the development of a targeted approach to sequencing the resistome. In this paper, the authors develop a primer set targeting 662 antibiotic resistance genes called DARTE-QM. They verify the bait set on a mock-community, and a variety of environmental samples. The authors provide valid statistical analysis. The quantitative aspect of this targeted approach is interesting and unique, as many other targeted approaches to studying the resistome are not quantitative (Guiton et al.). Unfortunately, the authors approach has several limitations. DARTE-QM failed to work in over half of the environmental samples tested, which is concerning for researchers. Furthermore, although the authors provided in vitro testing of their technique, they failed to validate it in silico. An in silico validation would help verify that DARTE-QM can capture all 662 antibiotic resistance genes, which the authors were unable to show in their in vitro approach. I recommend this

manuscript for publication with major revisions. Overall, I really enjoyed reading this manuscript and with a few changes I think this manuscript will be much stronger.

Here are my specific comments:

10. Introduction:

Lines 74: I suggest instead of 'scaling' to try 'large scale'.

Response:

Thank you, changed as suggested.

Line 74: "Despite being effective for the task, the cost per sample of employing metagenomic methods to elucidate resistomes often inhibits studies from large scale analyses."

11. Line 76: I suggest replacing "minute" with "rare" or small"

Response:

Thank you, changed as suggested.

Line 75: "This cost is driven by the necessity of having to indiscriminately sequence all DNA within an environment, of which the resistome often comprises only a fraction of a percent."

12. Lines 80-82: Another potential citation would be Beaudry et al., 2021 preprint on developing baits for AMR. Similar to the Guitor paper.

Response:

Thank you for suggesting this preprint, we added the citation to line 82 supporting the claim that enrichment of samples allows for increased information gained.

13. Results: Lines 103-105: How did the authors determine what ARGs to target? How did the authors obtain the ARG sequences? Please provide a reference or reference the specific section in the materials and methods.

Response:

Thank you. The ARGs were based on targeted a maximum number of genes from the ResFinder database and also abundant ARGs identified in previously published metagenomes. We added some text to clarify in the methods.

Line 455: "There were 2,472 sequences were obtained from the ResFinder database (version 3.2, November, 2016)²⁵, associated with 67 antibiotic resistance families. ResFinder was selected on account of its manual curation of genes associated with acquired antibiotic resistance. An

additional 409 ARG-associated sequences chosen as well, which had previously demonstrated high prevalence in animal agriculture based on their relative abundance in previously published metagenomes¹⁹.”

14. Lines 112-113: Can the authors explain what they mean by “true environmental resistomes”? Arguably, there is no such thing due to biases present in every step of the process (sampling, DNA extraction, PCR, sequencing, bioinformatics). This is why we use mock communities to establish ground truth. Please alter this sentence.

Response:

Thank you, and we agree that the use of ‘true’ here was extraneous. We have modified the sentence incorporating this suggestion and it now reads: “We also examined how DARTE-QM was able to characterize resistomes associated with manure, swine fecal, and agricultural soil samples (Supp. Table 4).”

15. Lines 121-123: It is very concerning that the authors had to throw out over half of their samples. It seems like this method will only work for high quality samples. Can the authors address this in the discussion? Were they able to try anything to get these samples to work?

Response:

Thank you for this comment. In response to this comment and the other reviewer, we have added details and discussed the samples that failed (see response to comment #3 and #4 above). We also note that not all manure-associated samples failed and it is unlikely that it is *just* the presence of manure-associated inhibitors that may be a challenge. More likely, it is a combination of both primer optimization and manure-associated inhibitors. We did not attempt to optimize for these samples within the scope of this effort, though that is our next step for a select set of genes.

16. Lines 129-137: I think it would be helpful if the authors made a table for this information.

Response:

This table is available in Supp Table 6.

17. Lines 150-154: Why are so many reads off-target?

Response:

In response to both reviewers, we have added more details in the discussion on the observation of off-target reads and sequencing artifacts. Please see Response above, #3 and #4.

18. Figures: Figure 2: I like the figure, however I think it is more intuitive to use a red-yellow-green scale rather than purple-green-yellow.

Response:

We agree that the red-yellow-green scale is aesthetically preferable, however, the color scale was chosen for being a well-established color-blind friendly palette to provide easier access to broader audiences.

19. Figure 3. For the barplots they are separated by sample type. Why include this information on the x-axis also? It makes it harder to read – just include the sample ID number.

Response:

Thank you, we have modified the figure as suggested.

20. Figure 4. Same comment as above regarding the x-axis.

Response:

Thank you, we have modified the figure as suggested.

21. Discussion: Lines 271-274: I think the authors need to be more specific here. In the materials and methods over half of the environmental samples did not work. That should be restated in the discussion, and this sentence should be altered to reflect that.

Response:

We have been modified the discussion significantly to better address the limitations of DARTE-QM in the discussion, specifically, addressing the environmental matrices effects, we have added text in the discussion. Please see responses #3 and #4 for specific text additions.

22. Lines 288-306: I think this paragraph should go third in the discussion. If you alter the last sentence of the second paragraph, like suggested above, it will flow nicely.

Response:

Based on both reviewers' comments, we have modified the discussion, hopefully for acceptable flow and clarity. Thank you for this suggestion, as we have incorporated into the current organization.

23. Other: One thing that I think is lacking is an in-silico simulation of the authors probe set. They are testing these in “real world” and mock communities, but do not have an understanding of if they all will work in ideal wet lab conditions.

Response:

The Beaudry paper that was recommended discusses an in-silico validation of their baits by aligning them to a reference genome and screening the downstream sequences to make sure they are getting the correct targets. In our study, we performed a similar method during our primer design. This approach, like the previously described in silico-validation, ensures that the primer

sequences are specific to capture the intended sequences, without overlapping with any others targets. For the mock community, we similarly screened for the ARGs that were present based on both our primers and our ARG database. To test the DARTE-QM primers in ideal wet lab conditions, we would suggest that each primer be tested separately to constrain for the impacts of multiple simultaneous amplifications. However, it is the high throughput nature of multiple probe and gene screens of DARTE-QM that is its benefit, thus we did not pursue individual experimental primer testing. Instead, as described below, we bioinformatically ensured that a ‘computational’ PCR would have adequately specific targets.

[Methods]: “The goal of primer design was to target the maximum number of our chosen sequences, with the highest specificity, staying within the set limit of 1,536 primers for the library preparation. Primers were designed using the Ribosomal Database Project’s EcoFunPrimer software: product minimum length = 220, product maximum length = 330, Oligo minimum size = 22, oligo maximum size = 30, maximum mismatch = 0, temperature minimum = 55, temperature maximum = 63, hair-pin max = 24, homo-max = 35, assaymax = 30, degenmax = 6, noTEendfileter = T, nopoly3GCfilter = T, polyrunfilter = 4, GCfilter min = 0.15 GCfilter max = 0.8. This produced 1,340 primers (Supp. Table 1) to target the ARG associated sequences, which accounted for 2,184 sequences (88.3%) from those selected (Supp. Table 2). This software tests primer sequences against reference sequences to ensure that targeted products are specific to ARGs of interest.”

“The Primer Design terminal program produces thermodynamically stable primer pairs for qPCR from an input of aligned nucleotide sequences.”

24. Could this approach be used in conjunction with other targeted approaches? Similar to how the authors suggest using it in approach with metagenomic libraries? Do the authors want to compare and contrast the results from their study to other published studies on targeted approaches to sequencing the resistome?

Response:

We focused our comparison to metagenomes and DARTE-QM because of the ability to obtain quantitative sequencing information on diverse genes. An alternative is to compare qPCR with subsequent sequencing of amplified products but because of the number of probes we include in DARTE-QM, this was considered not appropriate within the scope of this study. We do acknowledge that generally a comparison of specific gene targets by metagenome, qPCR, bait and capture, and DARTE-QM would be valuable but also recognize that the choice of the technology is very likely shaped by the research question. Another consideration in comparing across published studies is that we are aware of differences in extraction protocols and lab standards. Thus, in selecting our comparisons, we made the decision to use samples for which we knew DNA extraction was done at the same time for both DARTE-QM and metagenomes and similar extraction kits (though performed in different labs) could maintain consistency.

REVIEWERS' COMMENTS:

Reviewer #1 (Remarks to the Author):

Smith et al. have improved their previously submitted manuscript describing the DARTE-QM method. Overall, I am satisfied with the new manuscript. Rereading the paper led me to have an additional minor comment and two spelling/grammar suggestions

Comment:

Line 159: "Across all samples, an inverse linear relationship ($R^2 = 160\ 0.68$) (Supp. Figure 1) was observed between the number of reads which had a primer identified and the percentage of those reads that were artifacts."

This is surprising as the authors mention that the artifact reads have primers on their 5' end at the beginning of the paragraph. Does this imply that artifacts often lack a primer? If that's the case the beginning of the paragraph should be modified.

Spelling / grammar:

Line 47 "entitled" seems to have been copied over in error. I guess it should be removed.

Line 199: "For exploratory ..." I am not a native speaker but but "For the exploratory ..." sounds more correct to me.

Reviewer #2 (Remarks to the Author):

Dear Authors,

Thank you for addressing all of my concerns and comments. I think the alterations to the manuscripts have improved the overall quality. I recommend this manuscript for publication.

Response to reviewer comments:

Line 159: "Across all samples, an inverse linear relationship ($R^2 = 0.68$) (Supp. Figure 1) was observed between the number of reads which had a primer identified and the percentage of those reads that were artifacts."

This is surprising as the authors mention that the artifact reads have primers on their 5' end at the beginning of the paragraph. Does this imply that artifacts often lack a primer? If that's the case the beginning of the paragraph should be modified.

Response: We have modified the sentence slightly, "Across all samples, an inverse linear relationship ($R^2 = 0.68$) (Supp. Figure 1) was observed between the number of reads which had a primer identified and the percentage of those reads with primers that were identified as artifacts." The artifacts being referenced have primers on their 5' end.

Line 47: "entitled" seems to have been copied over in error. I guess it should be removed.

Response: Thank you, and we have removed this.

Line 199: "For exploratory ..." I am not a native speaker but "For the exploratory ..." sounds more correct to me.

Response: Thank you for this suggestion, we have considered the change but decided to leave it unchanged.